# Protein Expression in Metastatic Melanoma and the Link to Disease Presentation in a Range of Tumor Phenotypes

**DOI:** 10.3390/cancers12030767

**Published:** 2020-03-24

**Authors:** Yonghyo Kim, Jeovanis Gil, Indira Pla, Aniel Sanchez, Lazaro Hiram Betancourt, Boram Lee, Roger Appelqvist, Christian Ingvar, Lotta Lundgren, Håkan Olsson, Bo Baldetorp, Ho Jeong Kwon, Henriett Oskolás, Melinda Rezeli, Viktoria Doma, Sarolta Kárpáti, A. Marcell Szasz, István Balázs Németh, Johan Malm, György Marko-Varga

**Affiliations:** 1Clinical Protein Science & Imaging, Biomedical Centre, Department of Biomedical Engineering, Lund University, 221 84 Lund, Sweden; jeovanis.gil_valdes@med.lu.se (J.G.); indira.pla_parada@med.lu.se (I.P.); aniel.sanchez@med.lu.se (A.S.); lazaro.betancourt@med.lu.se (L.H.B.); boram.lee@med.lu.se (B.L.); roger.appelqvist@bme.lth.se (R.A.); henriett.kovacs-oskolas@med.lu.se (H.O.); melinda.rezeli@bme.lth.se (M.R.); szasz.attila_marcell@med.semmelweis-univ.hu (A.M.S.); johan.malm@med.lu.se (J.M.); gyorgy.marko-varga@bme.lth.se (G.M.-V.); 2Division of Oncology and Pathology, Department of Clinical Sciences Lund, Lund University, 221 85 Lund, Sweden; lotta.lundgren@med.lu.se (L.L.); hakan.olsson@med.lu.se (H.O.); bo.baldetorp@med.lu.se (B.B.); 3Section for Clinical Chemistry, Department of Translational Medicine, Lund University, Skåne University Hospital Malmö, 205 02 Malmö, Sweden; 4Department of Surgery, Clinical Sciences, Lund University, Skåne University Hospital Lund, 222 42 Lund, Sweden; christian.ingvar@med.lu.se; 5Chemical Genomics Global Research Lab, Department of Biotechnology, College of Life Science and Biotechnology, Yonsei University, Seoul 03722, Korea; kwonhj@yonsei.ac.kr; 6Department of Dermatology, Venereology and Dermatooncology, Semmelweis University, 1085 Budapest, Hungary; domaviki@gmail.com (V.D.); skarpati@t-online.hu (S.K.); 7Department of Bioinformatics, Semmelweis University, 1091 Budapest, Hungary; 8Department of Dermatology and Allergology, University of Szeged, H-6720 Szeged, Hungary; nemeth.istvan.balazs@med.u-szeged.hu; 9Department of Surgery, Tokyo Medical University, 6-7-1 Nishishinjiku Shinjiku-ku, Tokyo 160-0023, Japan

**Keywords:** metastatic melanoma, metastasis signaling pathways, BRAF mutation, mitochondrial function, clinical trials, targeted therapy, immunotherapy, therapeutic opportunities, clinical proteogenomics, histopathology, combinative treatments

## Abstract

Malignant melanoma is among the most aggressive skin cancers and it has among the highest metastatic potentials. Although surgery to remove the primary tumor is the gold standard treatment, once melanoma progresses and metastasizes to the lymph nodes and distal organs, i.e., metastatic melanoma (MM), the usual outcome is decreased survival. To improve survival rates and life span, advanced treatments have focused on the success of targeted therapies in the MAPK pathway that are based on BRAF (BRAF V600E) and MEK. The majority of patients with tumors that have higher expression of BRAF V600E show poorer prognosis than patients with a lower level of the mutated protein. Based on the molecular basis of melanoma, these findings are supported by distinct tumor phenotypes determined from differences in tumor heterogeneity and protein expression profiles. With these aspects in mind, continued challenges are to: (1) deconvolute the complexity and heterogeneity of MM; (2) identify the signaling pathways involved; and (3) determine protein expression to develop targeted therapies. Here, we provide an overview of the results from protein expression in MM and the link to disease presentation in a variety of tumor phenotypes and how these will overcome the challenges of clinical problems and suggest new promising approaches in metastatic melanoma and cancer therapy.

## 1. Introduction

Melanoma is a cancer that is considered the most aggressive skin cancer and has among the highest metastatic potentials of any malignancy worldwide [1,2]. Based on the GLOBOCAN database, 287,723 melanoma cases and 60,712 resultant deaths were reported in 2018. Worldwide, melanoma is thus ranked the 15th most common cancer [3,4]. In Sweden with a population of 10.3 million, over 4000 new cases are diagnosed annually and approximately 500 patients die from disseminated melanoma [5]. The absolute majority of patients diagnosed with malignant melanoma are cured with surgery, a radical excision of the primary tumor, and will have no further problems with recurrences and disseminated disease. With disseminated disease, the prognosis becomes worse and systemic treatment is not always successful due to the complexity and heterogeneity of the disease [6,7,8].

Understanding the complexity of melanoma is difficult, and many questions arise: What kind of pathobiological process initiates a metastasis-prone melanoma? What are the underlying tumor biological differences in a non-progressive versus a progressive malignant melanoma? What mechanisms provide a survival advantage to the lethal variants of MM? How can these tumors be detected? How can these patients be permanently cured? These questions form the cornerstones of the European Cancer Moonshot Center in Lund, Sweden and it is here that solutions to address these questions are being developed, research done with our mission statement ‘we conduct cancer research to end cancer as we know it, and to help benefit society as a whole’. We have a holistic view, where we cover melanomas from the entire body as illustrated in Figure 1. From melanoma tissues to a molecular level, including genome, transcriptome, and proteome, directly associated with melanoma disease signatures for disease presentation. Based on these features, upregulated metastatic mechanisms are driving uncontrolled proliferation and advanced metastasis showing highly heterogeneous cells in MM. 

Finding answers to these questions begins with interconnecting experiences from hospitals. Here clinicians interact directly with the patient and dedicate time to understanding the development of the cancer. This information can then be combined with the details obtained through pathological investigation including the morphology of the tumors, whether primary or metastatic. For these purposes, clinical data as well as tumor tissue and blood are collected at the clinic and the biological material stored in a biobank at the hospital. To suggest an all-encompassing solution for the treatment of MM patients, however, a more in-depth data repository needs to be generated. The complexity of the tumors and the variety of cells therein can then be integrated with proteogenomic expression data. The aim is to generate as much information as possible about the tumor by dissecting to the level of cellular representation, and further still, to the molecular representation of the cell contents and the major functions driven by the tumor. 

Various attempts to answer these questions include U.S. Food and Drug Administration (FDA)-approved treatments for MM with novel therapeutic agents and conventional therapies [9,10,11,12]. In particular, this applies to targeted therapy in patients where treatment selection is based on protein expression profiles related to the BRAF V600E mutation, MEK pathway, c-KIT pathway, PI3K pathway, and others [10,12]. From the available FDA-approved treatments, however, none of these options clearly and effectively manage MM. Once the tumor has spread from the primary source, metastatic melanoma rapidly acquires resistance and insensitivity to continuous treatments of targeted therapy within six to eight months [13,14]. Over the past several years, novel strategies through immune checkpoint blockade have been developed to target malignant melanoma and have started to transform the paradigm in melanoma care [14,15]. 

As protein expression induces a melanocyte transformation in MM, new therapeutic approaches have emerged based on cellular proteome profiles and the identification of molecular markers involved in the pathogenesis of transformation process in malignancy and metastases [12,16]. The various melanoma subtypes are dependent on origin, location and mutational profiles [17]. Disease presentation according to histopathological images, molecular mechanisms-of-action, protein expression profiles, and therapeutic opportunities for treatments are crucial requirements to generate practical road maps to link clinical evidence to solutions in the treatment of MM. 

In this perspective paper, protein expression in MM linked to disease presentation in a variety of tumor phenotypes is reviewed. The aim is to provide background information for improving interpretation of the disease and treatment of metastatic disease.

## 2. Disease Presentation of MM Incorporating Histopathology 

### 2.1. Disease Presentation with BRAFV600E in MM Tissue

As one of the representative regulators of cell growth and proliferation, BRAF is a key protein in the RAS-MAPK signaling pathway [18]. For > 50% of all melanomas and melanocytic nevi, the BRAF mutation with constitutive downstream activation has been reported [19]. The majority of the BRAF mutations are due to valine substitution (codon 600 from exon 15) [20]; and in approximately 75% of cases, the valine residue is replaced by glutamic acid (V600E). Other substitutions include valine to lysine (V600K) (~20%) and valine to arginine (V600R). This BRAF V600 mutation increases the kinase activation of BRAF and constitutively triggers the signaling cascade of RAS-RAF-MAPK [21]; thus promoting cell proliferation and metastatic activation [22]. In particular, the BRAF V600E mutation correlates highly with a poor prognosis and survival rate in melanoma patients [23]. 

For most of the cases where the patient has metastatic or non-resectable melanoma, various assays to determine the status of the mutated BRAF need to be performed to ascertain disease progression and subsequently, an appropriate treatment. Still considered as the gold standard in clinical practice, the most common methods to detect mutated BRAF expression at the molecular level are based on DNA assays (sequencing-based methods such as Sanger sequencing and pyrosequencing). In addition, an immunohistochemistry (IHC) approach that utilizes various monoclonal antibodies that are specific for the BRAF V600E mutation only (not wild-type or other epitopes of the BRAF V600 mutation) has been established for clinical diagnosis [24,25]. Providing rapid diagnosis at a low cost, this IHC method has shown high sensitivity, specificity, and availability across all tissues from melanoma patients [26]. The detection and quantitation of the mutated BRAF V600E protein in melanoma tissue by histopathological methods has some disadvantages. According to national guidelines, the required read-out from an assay for the BRAF mutation by classical immunohistochemistry and correlation with the correct targeted therapy are still highly variable.

In clinical practice, this histopathological characterization of melanoma tissue is correlated with disease presentation. The standardized immunohistochemical reaction is based on the identification of the mutated BRAF V600 protein by the VE1 clone (mutation-specific mouse monoclonal antibody) using an automated colorimetric visualization system [27]. Depending on the visualization method, the brown (diaminobenzidine, DAB) or red (alkaline phosphatase, ALP) staining of the tissue cells is deemed positive. At the tissue level, however, some samples display broad heterogeneity at the intercellular level and even in well-defined tissue compartments within the same tissue section (Figure 2). As specific parameters for BRAF V600E IHC have not yet been defined, tissue heterogeneity is neither included in routine histopathological evaluation nor in the prediction of responsiveness to targeted therapy. Nevertheless, precise tissue-based analysis on topographical expressional differences of the mutated BRAF protein can provide a basis for further translational clinical and pathological investigations in the future.

### 2.2. Comparison of Fresh Frozen and FFPE Metastatic Tumor Tissue Images Indicating the Associated Challenges of Histopathological Evaluation of MM

High resolution images of tumor tissue from formalin-fixed and paraffin-embedded (FFPE) (Figure 3a) and fresh frozen samples (Figure 3b–d) are obtained. Correctly processed tissue material from both frozen and FFPE tissue should be considered when analyzing disease presentation in MM tissues. 

For histological investigation and detailed morphological investigation, the images obtained from FFPE tumors are usually superior. This is due to the high resolution that can be obtained, and the extended timeframe where the tissues maintain morphological stability. To overcome the problems of protein cross-linking and denaturation by heat processing steps, multiple sampling strategies were recently applied to FFPE tissues (including heat-induced epitope retrieval, HIER) and combined with proteomics [28,29,30,31,32]. Despite these advancements, however, there are still some limitations with FFPE tissues, e.g., low protein quantities and lack of flexibility in the optimization of conditions for subsequent proteomic analysis. Conversely, fresh frozen material is in most cases preferred for protein identification and quantitation because the proteins in the tissues are still native and have not lost any activity or function. Nevertheless, only low-resolution IHC images can be obtained from fresh frozen tissue sections. In addition, these are extremely sensitivity to ambient temperature alterations and various other environmental influences. 

As such, the ideal combination is a systematic approach that integrates digital histopathology of FFPE tissues (c.f., Qupath software [33]) with proteomic data from LC-MS/MS analysis of fresh frozen tissues [27,34,35,36]. Based on protein markers in MM tissues, this combination will provide researchers and clinicians with new insights into the progression of melanoma (Figure 4). Furthermore, such an approach will offer novel clinical information that includes mechanistic variation and statistical analysis of the proteome to ultimately establish a clinical database that will improve the classification of MM patients.

## 3. Metastatic Mechanisms in Melanoma 

### 3.1. Targeted Therapies to Inhibit the Mutated BRAF Signaling Pathway in MM

Targeting BRAF V600E reduces both the activity and subsequent development of metastases by regulating IL-8 with cell extravasation [37] (Figure 5). The BRAF V600E mutation has also been implicated in the formation of new blood vessels by inducing secretion of VEGF (vascular endothelial growth factors) [38] and MIC-1 (macrophage inhibitory cytokine-1) [39]. Both are key factors in angiogenesis and proliferation in melanoma cells. 

The first BRAF inhibitor, sorafenib (Nexavar^®^, developed by Bayer Pharma AG, Berlin, Germany, and Onyx Pharmaceuticals, San Francisco, CA, USA) was developed for the treatment of melanoma. The drug is a pan-inhibitor of BRAF [40], however, sorafenib was not effective in treating melanoma. When administered alone; or as a combinatorial treatment with other chemotherapeutic, efficacy was low [41]. Assessment of other inhibitors in various clinical trials targeting BRAF mutations led to the development and approval by the Food and Drug Administration (FDA) of vemurafenib (Zelboraf^®^, Plexxikon, Berkeley, CA, USA, and Genentech, San Francisco, CA, USA) and later dabrafenib (Tafinlar^®^, GSK, London, UK). These BRAF inhibitors are available in several countries for the treatment of BRAF-mutated metastatic melanoma [42]. 

BRAF inhibitors are well-established compounds to target BRAF mutations in melanoma patients. Major concerns, however, are related to the development of drug resistance that is directly connected to recurring melanoma following inhibitor treatment. Several studies have suggested that drug resistance in melanoma occurs by reactivation of the MAPK pathways and ERK1/2 activity [43]. These pathways directly regulate proliferation, invasion, and metastases in melanoma and increased cell survival rates. Approximately 40–60% of cutaneous melanoma cases have reported that the BRAF mutation was induced through activation of the MAPK pathways [44,45]. Over the past few decades, patient management has developed and improved through targeted therapies that are directed towards MAPK pathways. For the BRAF V600E mutation in melanoma, patient therapy has focused on the MAPK pathways that primarily involve BRAF and MEK. These include single (vemurafenib or dabrafenib) or combinatorial treatments with trametinib (Mekinist^®^, GSK, London, UK) and cobimetinib (Cotellic^®^, Exelixis, San Francisco, CA, USA, and Genentech, San Francisco, CA, USA) [46,47,48]. Such strategies have revealed significant clinical advantages such as decreased tumor size in melanoma patients. Nevertheless, efficacy is still under question because of the diverse responses observed in patients. In the majority of cases, after an initial period of tumor regression, drug resistance following a drug-free period occurs, and the disease progresses [49]. 

### 3.2. The Key Role of Mitochondria in MM

Multiple studies have suggested that the inherent metabolism of tumors actually play a key role in cancer development and metastasis formation; including resistance to many therapies [50,51]. In addition, tumor signaling pathways that are regulated by BRAF, HIF-1α, c-MYC, and mTOR are accelerated to activate glycolysis in the cancer cells and accumulation of lactate in the microenvironment of the tumor [52]. Through dysregulation of glycolysis, tumor cells promote abnormal growth, metastases, and recurrence of the disease [53,54].

During the progression of malignant melanoma, the up-regulation of glycolytic enzymes (e.g., the hexokinases, and isoform M2 of the pyruvate kinase, PKM) together with the first enzyme in the gluconeogenesis pathway (phosphoenolpyruvate carboxykinase [GTP], PCK2) suggests that melanocytes undergo a metabolic shift towards a glycolytic phenotype [55]. More recently, our group and others have observed that mitochondrial oxidative phosphorylation (OXPHOS) up-regulation is a common characteristic in metastatic melanoma and tumors that developed resistance to targeted therapy [56]. It is well known that the BRAF mutations present in approximately 50% of melanomas trigger metabolic reprograming from oxidative phosphorylation to glycolysis [57]. Following targeted treatment, tumor cells that develop resistance experience up-regulation of OXPHOS. In addition, mitochondrial translational proteins have also been observed as up-regulated during melanoma progression. Altogether, these findings highlight the important role of mitochondria in melanoma. Subsequently, mitochondrial proteins and pathways can be considered as potential targets; particularly for tumors that are resistant to targeted therapy. 

In support of this hypothesis, a positive correlation has been observed between the proliferation rate of tumor cells and the levels of OXPHOS and mitochondrial translational proteins. This confirms that for tumors that are resistant to targeted therapy and; regardless of the BRAF mutational status, also highly proliferating cells have the potential to be treated by a targeted mitochondrial therapy [58]. 

Endorsing this theory, a recently published study analyzed the differences between responder and non-responder metastatic melanoma patients to immunotherapy. The results indicated that oxidative phosphorylation was up-regulated in responders [59]. Taking these findings into account, it is envisioned that mitochondria play a key role in malignant melanoma. Highly proliferating melanoma cells, and cells resistant to targeted therapy both up-regulate mitochondrial proteins. Interestingly, higher levels of mitochondrial proteins improve the response to immunotherapy. Two therapeutic opportunities thus arise from the analysis of mitochondrial proteins in metastatic melanoma; either immunotherapy when possible, or direct targeting of mitochondrial function. 

### 3.3. Drug-Treatment Impacts from Protein Expression in Melanoma Patients

Despite current diagnostic and treatment regimens, advanced melanoma in stage III/IV still presents enormous therapeutic challenges with a short survival rate of less than 1 year [60]. As shown in a phase III clinical study [43], the median progression-free survival rate is only 6 months. 

From recent research, several molecular protein markers are already used to monitor progression and relapse, e.g., S100B, MART1 and PMEL, and S100A13 [61]. Nevertheless, not only the evaluation of molecular protein markers is required. In addition, more detailed studies that analyze the relevance of the metastatic characteristics of melanoma, clinical outcome, and selection of optimal treatment strategies still need to be established [62]. 

There is a great demand for validated biomarkers that not only support the primary diagnosis; but can also predict the progression of the disease and response to the treatment of metastases. Accordingly, the need for routine clinical biomarkers to monitor disease progression and treatment efficacy is emerging with high importance.

Protein expression profiling is crucial to understand the disease presentation of melanoma. As such, focus on protein markers to monitor tumor progression and the tumor environment in MM is key. Protein profiling studies that involve mass spectrometry-based proteomics have been utilized as an efficient and broad technique to analyze and evaluate the regulation of proteins under various conditions, elucidate molecular mechanisms, and determine the status of protein networks in MM.

From unpublished results on protein expression profiling by proteomics that utilized high-resolution mass spectrometry and bioinformatics, our group evaluated changes that can occur at the protein level between patients that were treated clinical treatments pre-operatively and untreated groups as shown in Table 1. Taking into account the inherent heterogeneity that exists in melanomas, patients’ samples (89) were optimally selected to provide the highest degree of variation that had been induced by the different treatments. A multi-factor analysis (MFA) [63] was performed on two groups of variables: (1) clinical parameters (treatment, age at diagnosis, gender, disease stage); and (2) histopathology (tumor cell content, connective tissue content, pigmentation score, necrosis content, lymphatic score, lymphatic distribution, lymphocyte density, adjacent lymphatic node area, predominant cytoplasm). The distribution of patients is displayed as a 2D plot (Figure 6a). Six untreated patients, indicated by the green arrows in the yellow ellipse, were chosen for comparison. A description of the selected patients is provided in Table 1.

The multiple linear regression (MLR) analysis detected nine differentially expressed proteins (DEP) between the treated and untreated patients (*p*-value <0.05) (Table 2). Four of these proteins (SRSF3, PLG, FGG, C3, SERPINA1) were previously described [64] as related to survival in patients with melanoma cancer. Interestingly, SRSF3, which was up-regulated in treated patients, showed previously a high expression in long survival patients, and conversely PLG, FGG, C3, SERPINA1 which were downregulated in treated patients (Figure 6b) showed high expression in patients with short survival. 

These finding led us to conclude that apparently the treatments lead to a long survival by increasing SRSF3 intensity and decreasing PLG, FGG, C3, SERPINA1 (Figure 6c). Similarly, CMPK1 and ABRACL were also upregulated in the treated group of melanoma patients. Conversely, high expression of HBG1 and HBG2 was shown in untreated patients. 

In this study, the protein expression profiles obtained from metastatic melanoma tissues that had been analyzed by high-resolution mass spectrometry and bioinformatics were further compared by a combined MFA and MRL analyses. These results were highly congruent with the proteomic analyses; including correlation with survival. It is envisaged that the clinical information obtained from patient metastatic tissue will provide novel protein markers that are related to the effects of treatment. This knowledge will hopefully lead to new strategies to evaluate melanoma progression and survival. Furthermore, our studies will aid in establishing efficient clinical approaches.

### 3.4. Immunotherapy with Checkpoint Inhibitors

In contrast to many other cancer types chemotherapy is now rarely used as first line treatment for malignant melanoma due to the poor efficacy, side effects and the fact that more effective treatment options now are available [42]. Immunotherapy is the favored treatment to induce the immune response of a patient to target and more effectively eliminate cancer cells. Several types of immunotherapy can be used to treat melanoma; and for approximately a decade now, many immunotherapies have been used in patients. In particular, immunotherapy that involves interferon and interleukin has been applied to treat advanced melanoma. Moderate clinical advantages and considerable toxicity, however, are emerging as a major limitation of this strategy [65]. Approaches to decipher the underlying mechanisms and molecules involved in immunoregulation are continuously emerging. With the advent of novel immunotherapeutic treatments that include checkpoint inhibitors, the cell-medicated immune system becomes activated to produce antibodies to target immune reactive molecules. Undoubtedly, with the inclusion of checkpoint inhibitors, immunotherapy is proving to be one of the most effective recent advances in cancer therapy [66]. Immunotherapy is now first line treatment for patients with metastatic disease and in many cases a PD-1 antibody gives good results. Cytotoxic T-lymphocyte antigen 4 (CTLA-4) antibodies are used for tumors with low PD1 expression and tyrosine kinase inhibitors only to BRAF-positive tumors. 

To activate the effective triggering response of a T cell, tumor antigens are presented as ligands to the T cell receptor (TCR). These then bind to the major histocompatibility complex (MHC) on an antigen presenting cell (APC). Next, a co-stimulatory reaction results in the binding of CD80/CD86 to CD28 on the lymphocyte. Through this step of T cell activation, an integrated immune reaction of innate and adaptive immunity is induced to eliminate malignant melanocytes [67]. Beginning with the development of antibodies, these approaches have shown promising results including inhibitory strategies to target the CTLA-4 and programmed death-1/programmed death ligand-1 (PD-1/PD-L1) [68]. Monotherapy with these checkpoint inhibition, however, primarily led to clinical failure and tumor regression [69]. To overcome this, Wolchok et al. performed clinical studies to determine the synergistic effect of combining anti-PD-1 (nivolumab) and anti-CTLA-4 (ipilimumab). The study demonstrated that the combined therapy significantly increased the positive clinical effects on advanced melanoma [69,70]. Similarly, the current standard for melanoma care that utilizes combinatorial immunotherapy has also shown applicability in advanced melanoma treatment [71,72]. For patients with advanced melanoma, immune checkpoint inhibitors are also actively utilized in combination with other targeted therapies, e.g., BRAF and MEK inhibitors [73]. 

Representing the newest and state-of-the-art therapy, a double combination treatment with checkpoint inhibitors has been reported for advanced melanoma. The outcome of a five-year trial (NCT01844505) by Larkin et al. suggested that the combination of nivolumab plus ipilimumab; or monotherapy with nivolumab showed more promising results with respect to PFS (progression-free survival) and OS (long-term overall survival) than monotherapy with ipilimumab. In addition, the health-related quality of life for the patients was not reduced [74]. 

Recently, a study presented promising results that TLS (tertiary lymphoid structures) from advanced melanoma patients, who were treated with checkpoint inhibitors, have important roles for maintaining immune response on microenvironment in melanoma through accompanying distinct patterns of T cell phenotypes. These results strongly suggested evidences that TLS formation could provide therapeutic strategies with improved clinical outcomes and immune responses related to cancer microenvironments [75]. 

## 4. Therapeutic Opportunities from New Approaches for MM 

### 4.1. Drug Resistance in MM

Newly developed drugs that enable targeted therapy, e.g., protein kinase inhibitors or drugs that modulate the immune response, have shown impressive initial results [76]. Following a phase of initial response, unfortunately, single-agent therapies for advanced cancer are rarely curative. This is due to the rapid development of resistance. 

Recently, several drug resistance mechanisms have been identified in melanoma treated with RAF inhibitors. These include genetic (MEK1 mutation [77,78], NRAS mutation [79], bypass activation of MAPK through receptor tyrosine kinases (RTKs, e.g., EGFR) [80,81], NF1 loss from RAF inhibition [82], and BRAF amplification [83]). Also non-genetic mechanisms e.g. COT activation [84], upregulation of EGF receptor/SRC family kinase signaling [85], PTEN loss with BIM suppression [86], alternative splicing of BRAF [87], and feedback inhibition of mitogenic signaling [88]) have been described. Moreover, increased expression and dysregulation of RTKs have also been reported as mechanisms for RAF inhibitor resistance [89,90]. 

Finally, resistance after treatment with checkpoint inhibitors has been reported including primary resistance to PD-1/PD-L1 or CTLA-4 from the lack of interferon gamma (IFNγ) signaling in the tumors [91,92,93].

Especially, after treatment with BRAF inhibitors, over 50% of the patients show re-progression and resistance within several months. Through the rapid recovery of the MAPK pathway, resistance to the BRAF inhibitor is quickly acquired and leads to clinical re-progression [94]. Most BRAF-resistant melanomas also involve additional mutations in the MAPK pathways, e.g., MEK1 mutations and BRAF amplification [42,95].

### 4.2. Combinatorial Treatment and Promising Therapies in MM

#### 4.2.1. Triple Combination of Checkpoint Inhibitor with Therapies Targeting MEK and BRAF

To overcome the co-activation of MEK in BRAF-mutated melanoma observed with monotherapy, combinatorial approaches using BRAF and MEK inhibitors have been utilized to improve patient survival rates [96]. Compared to a BRAF inhibitor alone, combined treatment with MEK inhibitors (e.g., Trametinib) and BRAF inhibitors (e.g., Dabrafenib) has resulted in increased PFS and OS [46,47,97]. Nevertheless, these combinatorial treatments are not able to prevent all the problems that can occur from disease recurrence, progression and metastasis [79]. 

Checkpoint inhibitors induce an overall low response rate and, on occasion, adverse effects related to immunity. Nevertheless, these inhibitors have showed promising therapeutic outcomes with long response times [98]. Some preclinical models have also been reported to improve anti-tumor activity. These include the combinatorial treatment of BRAF and MEK inhibitors with anti PD-1 inhibitors that led to an extended period of response [99,100,101]. 

Cornerstone research recently revealed the emergence of triple combination treatments that combine immune checkpoint inhibitors and targeted therapies that resulted in promising efficacies in advanced melanoma [73,102,103]. Ribas et al. reported a clinical trial (NCT02130466) of BRAFV600-mutated metastatic melanoma patients. The results suggested an improvement in anti-tumor activity through a triple therapy combination of dabrafenib (BRAF inhibitor), trametinib (MEK inhibitor), and pembrolizumab (anti-PD-1 antibody). This evaluation reported that the triple therapy combination increased the frequency of long-lasting responses; and as such, could also be a suitable treatment for patients with poor prognosis to monotherapy [102]. In a parallel study (NCT02130466), Ascierto et al. reported a phase 2 trial (including a randomized double-blind placebo) that showed valuable results with encouraging PFS. The triple therapy combination of dabrafenib (BRAF inhibitor), trametinib (MEK inhibitor), and pembrolizumab (anti PD-1 antibody) were administered to patients with advanced BRAFV600-mutated metastatic melanoma [73]. As a clinical report from a phase Ib study (NCT01656642), Sullivan et al. evaluated a triple combinatorial therapy of atezolizumab (anti-PD-L1 antibody), cobimetinib (MEK inhibitor), and vemurafenib (BRAF inhibitor) in patients with BRAFV600-mutated metastatic melanoma. From this evaluation, the triple combination showed promise with anti-tumor activity [103]. Although a long-term benefit for triple treatments still needs to be fully evaluated, to our knowledge, these three published studies show encouraging results. Particularly, when compared to the early reports of BRAF and MEK inhibition. These triple combinations may be applicable to treat patients who show poor prognosis to first-line targeted therapy and urgent cases where the patient has limited time to wait for a response to a checkpoint inhibitor [104]. 

Lately, engineered oncolytic viruses have been shown to be effective for local and systemic immunotherapies in cancer. A few years ago, the first oncolytic virus was approved by the FDA for melanoma immunotherapy [105,106,107,108]. This therapy is used not only for non-resectable skin and lymph node lesions but also for melanoma metastases elsewhere [109]. Recurrent, unrespectable stage IIIB-IVM1a melanoma patients have been treated with talimogene laherparepvec (T-VEC), the oncolytic virus, which is the first approved therapy [105].

The conclusion of these reports is that with new insights into diagnosis and management of MM and the subsequent deconvolution of problems that require further management; patients will receive improved treatments.

#### 4.2.2. Combination of the HDAC Inhibitor on MM

Histone deacetylase (HDAC) proteins are a family of epigenetically regulated enzymes that remove acetyl groups from lysine residues in histones. Through regulation of the structure of chromatin and transcriptional factors of several genes that are linked to cell division and cell cycle, HDACs can be directly correlated with malignant characteristics of tumors [110,111]. HDACs can also regulate genes by modulating the various targets of histone and other non-histone proteins, e.g., signal mediators, DNA enzymes, and various chaperones [112]. By regulating HDAC-mediated deacetylation, HDAC inhibitors are thus emerging as a promising therapy to inhibit tumor cell proliferation and environmental factors thereof [76,113]. 

At present, the perspectives for forthcoming and promising treatment for melanoma include various reported studies on the combination of HDAC inhibitors with not only targeted therapy; but also with checkpoint inhibitors. In particular, combinatorial checkpoint inhibitors with HDAC inhibitors has shown significant results with respect to immune-responsive activities on tumor cells in vitro and in vivo [114,115]. This includes an increase in immune response [116] and augmented immunotherapy via PD-1 blockade in melanoma [113]. 

According to prior evidence and success with HDAC inhibition, various ongoing clinical trials are underway. These include trials assessing combinatorial checkpoint inhibitors with HDAC inhibitors. A clinical evaluation that combined pembrolizumab (anti PD-1 antibody) with entinostat (HDAC inhibitor) is in progress for uveal MM (NCT02697630) [117]. At the H. Lee Moffitt Cancer Center and the Novartis Research Institute, a phase I clinical trial in advanced and unresectable melanoma patients has been performed. Here, ipilimumab (anti CTLA-4 antibody) was combined with panobinostat (NCT02032810) [112]. Data from a phase Ib/II trial of ENCORE-601 (NCT02437136) in metastatic and progressed melanoma patients showed that when significant resistance was acquired following prior PD-1 blockade or prior PD-1/PC-L1 blockade, a combinatorial therapy with entinostat and pembrolizumab showed remarkable results and increased anti-tumor activity [112,118,119]. As another clinically applicable example of combinatorial targeted therapy, Maertens et al. treated melanoma models harboring mutations (BRAF, NRAS, and NF1) with entinostat and BRAF/MEK inhibitors. The study demonstrated that the drug combination suppressed gene expression related to DNA repair and enhanced destruction of the melanoma cells [120]. 

In conclusion, the preclinical data obtained from these studies still provide insufficient information for therapeutic indications; albeit, efficacy and toxicity information of HDAC inhibitors is given. Further clinical trials that integrate advances in epigenetics will expedite clinical optimization to reach the ultimate goal of treating advanced melanoma patients with new approaches involving HDAC inhibition and various combinations thereof (Table 3).

#### 4.2.3. New Effective Approaches to Evaluate MM

In the context of disease, tumor progression involves metabolic changes and dysfunction of cellular processes that lead to the pathological progression [121]. The tumor proteome represents a particular metabolic stage and is a dynamic entity that varies during tumor progression. Therefore, proteomic research to study proteome dynamics in melanoma would be highly-valuable in understanding tumor progression and development of the disease [122]. During the steps of tumor progression, various changes in the proteome of the cancer cells induce abnormal growth and interrupt host homeostasis. To elucidate the interplay between pathological processes and the biological events that lead to cancer progression, it is imperative to evaluate these proteomic changes [123]. 

Mass spectrometry (MS)-based proteomics is a powerful approach to identify and quantitate proteins in biological samples [124]. The abundance of proteins has important biological significance; and determining protein expression levels, post-translational modifications and protein interactions in the context of disease processes is of high importance. In recent years, several MS-based proteomic methods have emerged including the ‘shotgun’ or gel-free approach that is applied in discovery proteomics. Proteins are enzymatically digested into peptides that are separated by liquid chromatography (LC) and analyzed by tandem mass spectrometry (MSMS). Peptides are identified and correlated with corresponding proteins by matching generated MSMS spectra with theoretical spectra from a protein sequence database [124]. To evaluate the mechanisms of MM at the molecular level continuously, proteogenomic-based strategies to explore novel and more specific MM proteins as disease markers are still urgently required. 

To overcome problems and identify solutions for MM, a proteogenomic study that incorporates deep proteome profiling has been performed (Figure 7). Although some of the driver gene mutations have been identified, these cannot act as individual indicators for every melanoma case. There is still a lack of evidence concerning the influence of heterogeneity in melanoma tissues and cells and the subsequent effect and outcome of targeted therapy. 

From a quantitative proteogenomic approach on tissue samples from melanoma patients, our group has attempted to analyze the proteins that were identified. The processed samples from patient tissues were from various sites, including normal skin, primary tumor lesions, tumor environment, local recurrence lesions, and metastasis (subcutaneous, lymph node, and distant metastasis). In the analyzed results, the metabolic reprogramming during carcinogenesis and progression of melanoma included a distinct shift in the function of the mitochondria and various biogenesis pathways. Based on these proteomic results, we are pursuing the discovery of dysregulated protein pathways and the genomic status in MM. These can be evaluated with Multi-OMICS, an in vitro cell-based study, and in vivo PDX studies. Especially, we have evaluated and identified key proteins regulating epigenetic pathways by modulating calcium-homeostasis (unpublished data). In addition, some of the proteins and the associated pathways that are involved in the dysregulated mechanisms have high potential as therapeutic targets for melanoma; particularly those that are resistant to targeted therapy. By demonstrating the mechanistic mode-of-action, these results suggest that deep proteogenome profiling via mass spectrometry and genomic analysis can reveal insights into how melanoma progresses and gives rise to metastases in patients.

## 5. Conclusions

The ultimate goal in the treatment of MM after surgical removal of the primary tumor is to prevent metastatic spread and treat disseminated disease with as few adverse events as possible. To achieve this goal, we need to fully understand the underlying molecular mechanism(s) and associated signaling pathways. 

Most of the current clinical trials on combinatorial therapy with checkpoint inhibitors and targeted therapy drugs have suggested high efficacies with respect to PFS and OS in MM patients. The options and choices for MM patients, however, are still limited because of the small number of drugs approved for clinical use and insufficient clinical data on long-term assessment. 

The perspectives and overview of MM, as discussed in this review, and the link to disease presentation still require continued attention. Approaches encompassing personalized medicine (or precision medicine) will assist in the creation of new clinical avenues for MM and cancer therapy. The tailoring of the individual needs of a patient must be based on tumor heterogeneity, mechanistic factors, and include systemic profiling of the status of the disease in the patient. In the last decade, personalized treatment has shown enormous advances that have resulted in remarkable achievements and improved outcomes in melanoma treatment. Evaluation of these experiences, reveals that these advances have taken into consideration the various clinical statuses with adverse or resistant effects, clinical risks, prediction of treatment, experience from treatments including interventions, medical background, and other clinical information [125]. Also, these approaches should not only be based on the fundamental knowledge of the molecular mechanisms related to the disease process; but also on the appropriate updates obtained from recent clinical research [126]. As the number of combinatorial treatments in clinical trials continues to rise, specific demands on personalized medicine will naturally define the direction of using such personalized strategies to include precise information obtained from MM patients.

## Figures and Tables

**Figure 1 cancers-12-00767-f001:**
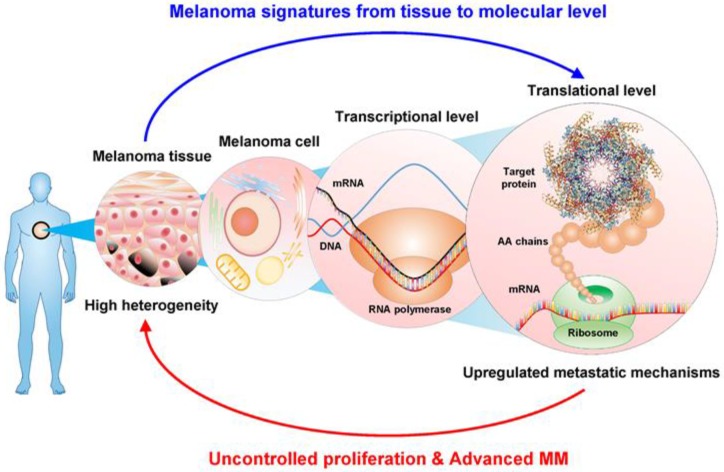
Holistic view of melanoma signatures for disease presentation from entire body to molecular level of cells.

**Figure 2 cancers-12-00767-f002:**
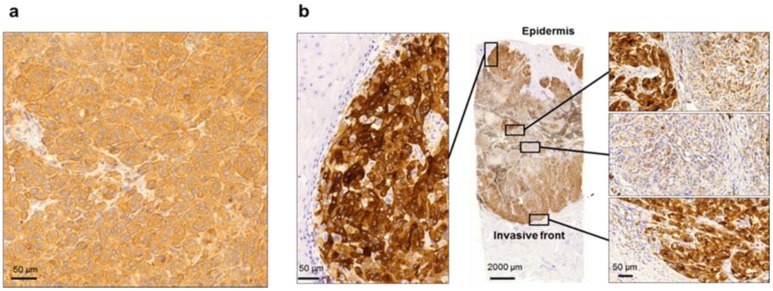
Expression of BRAF V600E mutation in melanoma tissue. With routine immunohistochemistry, the distribution of BRAF V600E mutation can be easily detected in melanoma tissue by a standardized colorimetric assay. Mutated BRAF V600E proteins react with HRP-DAB (peroxidase conversion of diaminobenzidine) (**a**) Homogeneity of the BRAF V600E protein expression. (**b**) Heterogeneity of the BRAF V600E protein expression. The peripheral and focal central regions of show a pronounced expression of the mutated protein. In areas with mild V600E expression, there is marked intercellular heterogeneity. BRAF V600E IHC, clone VE1, HIER (heat-induced epitope retrieval) pH = 9; OM 112 ×.

**Figure 3 cancers-12-00767-f003:**
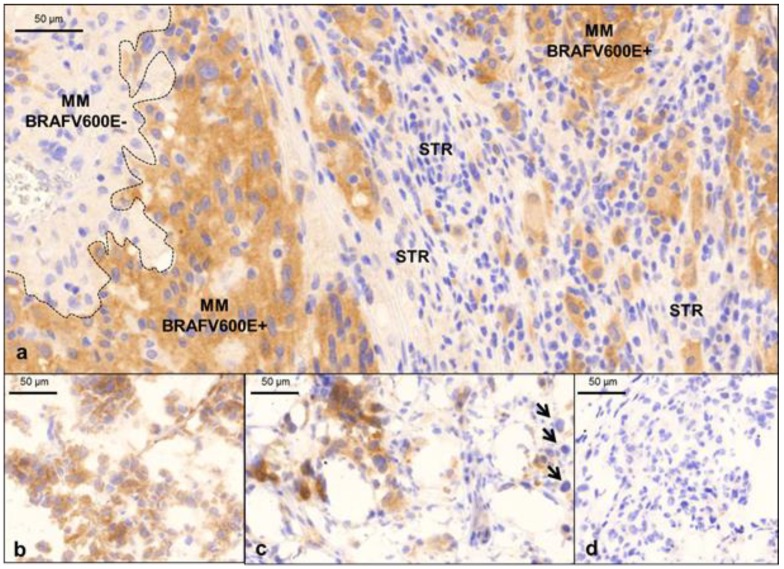
IHC examples obtained from melanoma patients. (**a**) Note the intratumoral heterogeneity of the FFPE tissue sample. The BRAF V600E-mutated melanoma cells display no immunoexpression of the mutated protein (left of the dashed line); whereas strong immunoexpression of the mutated BRAF protein is evident to the right of the dashed line. Stromal inflammatory cells (STR) are the negative, internal controls for staining. (**b**) Homogenous patterns of expression are shown in cryosections of fresh frozen BRAF V600E-mutated melanoma samples. (**c**) Heterogeneous patterns of expression are shown in cryosections of fresh frozen BRAF V600E-mutated melanoma samples (arrows indicate melanoma cells that are negative for the mutated protein). (**d**) A fresh frozen wild-type melanoma sample is the negative control for the IHC method.

**Figure 4 cancers-12-00767-f004:**
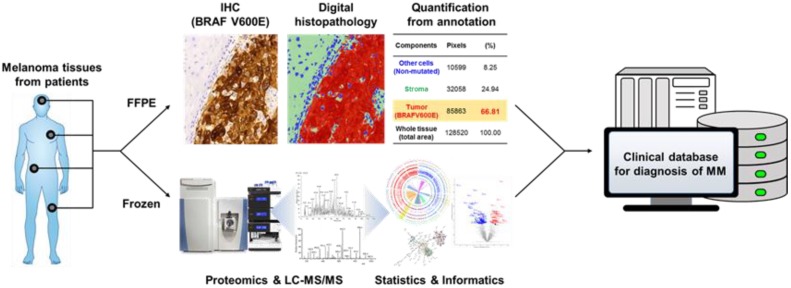
A systematic approach for melanoma tissue that combines digital histopathology and proteomics.

**Figure 5 cancers-12-00767-f005:**
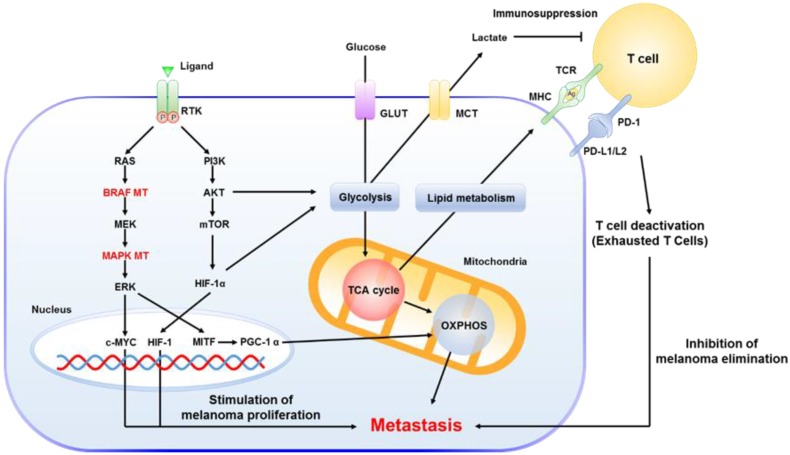
Metastatic melanoma. An overview of the mechanisms involved in regulation of metastases by stimulation of downstream signaling pathways through RTK activation, immunosuppression of T cells, and mitochondrial function.

**Figure 6 cancers-12-00767-f006:**
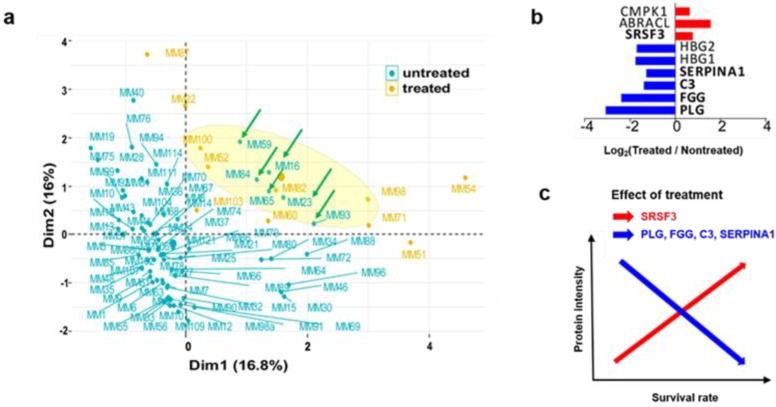
Drug treatment effects on protein expression in MM. (**a**) Multiple factor analysis based on clinical and histopathological information. Dots represent patients. The axes Dim1 and 2 represent the first two principal components that consider the contribution of all active groups of variables to define the distance between individuals (patients). The ellipse was generated at a 99% confidence interval. Green arrows indicate untreated patients selected to be compared with the treated patients (yellow) in further analyses. (**b**) Differentially expressed proteins between treated and untreated patients. Only proteins significantly associated (multiple linear regression, *p*-value <0.05) with the variable treatment were considered relevant. (**c**) Association between protein intensity and survival rate in patients with melanoma [64]. The treatment leads to a long survival by increasing SRSF3 and decreasing PLG, FGG, C3, SERPINA1.

**Figure 7 cancers-12-00767-f007:**
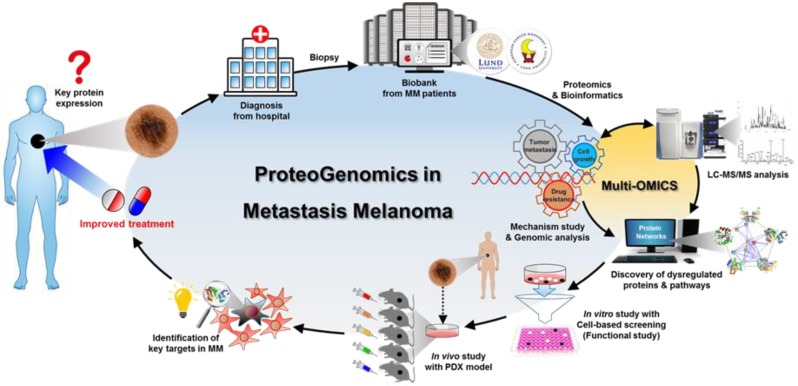
Workflow of proteogenomics in MM. Patient samples are processed by deep proteome profiling via a Multi-OMICS approach to reveal insights into the action of metastatic melanoma.

**Table 1 cancers-12-00767-t001:** Clinical and histopathological information of samples compared by linear regression.

Categories	Clinical Information				Histopathological Information (Average)								
Code	Age of Diagnosis	Gender	Disease Stage	Treatment	Connective Tissue Content	Lymphatic Score	Tumor Cells Content	Adjacent Lymph Node Area	Pigmentation Score	Predominant Cytoplasm	Lymphatic Distribution	Necrosis Content	Lymphocyte Density
MM16	55	Female	4	Nontreated	9.8	3	53.5	1.8	3	1	2	35	1
MM23	71	Female	4	Nontreated	19	3.7	50.3	30.7	1	2	1.7	0	2
MM59	38	Female	4	Nontreated	12.5	2	62	0	2	2	1	25.5	1
MM65	59	Female	4	Nontreated	11.7	2	67.3	0	1	2	1	21	1
MM84	52	Female	4	Nontreated	1.3	2	71.3	0	1.3	1	1	27.3	1
MM93	70	Male	4	Nontreated	83	3	15	2	1	0	1	0	2
MM87	29	Female	3	BRAF	7	3	62	30	0	1	1	1	2
MM100	64	Female	3	BRAF	8	2	80	5	0	1	1	7	1
MM103	61	Male	3	BRAF	2	2	98	0	0	2	1	0	1
MM60	77	Female	3	Chemotherapy	2	0.7	98	0	0.7	2	0.3	0	0.3
MM71	61	Male	4	Chemotherapy	7.7	3	92.3	0	3	2	1	0	2
MM82	76	Female	3	Chemotherapy	13.3	2	65	16.7	0	1	1	5	1
MM98	55	Male	4	Chemotherapy	17	4	75	0	2	0	2	8	2
MM22	44	Male	3	Interferon	5	5	25.7	69	3	2	3	0.3	2
MM52	68	Female	3	Interferon	7	2	93	0	1	1	1	0	1
MM54	81	Female	4	Interferon + Chemotherapy	6.3	1.3	57.7	0	0	2	0.7	36	0.7

**Table 2 cancers-12-00767-t002:** Differentially expressed proteins between patients with neoadjuvant therapy and no preoperative treatment.

Accession	Gene	Log2.Fold Change	*p*-Value			
			Treatment	Age of Diagnosis	Gender	Disease Stage
P00747	PLG *	−3.0721709	0.033233	0.083008	0.511929	0.576933
P02679	FGG *	−2.3908585	0.014045	0.269768	0.979767	0.189957
P01024	C3 *	−1.389545	0.048588	0.328285	0.96424	0.195198
P01009	SERPINA1 *	−1.2769647	0.027569	0.970118	0.84077	0.07742
P84103	SRSF3 *	0.779147	0.049004	0.119298	0.649564	0.653696
P69891	HBG1	−1.7589103	0.02118	0.352096	0.215719	0.071367
P69892	HBG2	−1.7018783	0.031238	0.445556	0.295623	0.100417
Q9P1F3	ABRACL	1.5688169	0.031732	0.631799	0.154456	0.077132
P30085	CMPK1	0.64424732	0.011524	0.131597	0.546014	0.062366

* These proteins previously associated to survival in patients with melanoma [64].

**Table 3 cancers-12-00767-t003:** Combinative treatments for metastatic melanoma.

Category	Applied Treatments for Melanoma Patients	Details (Clinical Trial ID)	References
Double combination	Trametinib (MEKi) Dabrafenib (BRAFi)	Improvement of PFS and OS than BRAFi as alone	[46,47,48,97]
	Nivolumab (anti PD-1 Ab) Ipilimumab (anti CTLA-4 Ab)	Improvements of PFA and OS than ipilimumab (NCT01844505)	[74]
Triple combination	Dabrafenib (BRAFi) Trametinib (MEKi) Pembrolizumab (anti PD-1 Ab)	BRAFV600-mutated metastatic melanoma patients (NCT02130466)	[102]
	Dabrafenib (BRAFi) Trametinib (MEKi) Pembrolizumab (anti PD-1 Ab)	Phase 2, randomized double-blind placebo-control (NCT02130466)	[73]
	Cobimetinib (MEKi) Vemurafenib (BRAFi) Atezolizumab (anti-PD-L1 Ab)	Phase Ib, BRAFV600-mutated metastatic melanoma patients (NCT01656642)	[103]
HDACi combination	Pembrolizumab (anti PD-1 Ab) Entinostat (HDACi)	Progression to metastatic melanoma on eye (NCT02697630)	[117]
	Ipilimumab (anti CTLA-4 Ab) Panobinostat (HDACi)	Advanced and unresectable melanoma patients (NCT02032810)	[112]
	Pembrolizumab (anti PD-1 Ab) Entinostat (HDACi)	Melanoma patients, showing significant resistances on prior PD-1 blockade or prior PD-1/CTLA-4 blockade (NCT02437136)	[118,119]
	Dabrafenib (BRAFi) Trametinib (MEKi) Entinostat (HDACi)	On treating melanoma models of harboring mutations of BRAF, NRAS, and NF1	[120]
	Anti PD-1 ab (from BioXCell) LBH589 (HDACi)	Slower tumor progression and increased survival compared with control and single treatments in B16F10 mice model	[113]
	Anti-PD-1 ab (from BioXCell) AR42 (HDACi)	Enhanced immunotherapy response with HDACi in B16 mice model	[116]

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
