# Peer review of "Protein Expression in Metastatic Melanoma and the Link to Disease Presentation in a Range of Tumor Phenotypes"

_cancers, 2020, doi:10.3390/cancers12030767_

Round 1

Reviewer 1 Report

The authors review the molecular basis of malignant melanoma, focusing on BRAF/MEK and PDL1 pathways of therapeutic interest. Overlap with metabolic and protein biomarkers is made. The manuscript would be greatly improved if additional information on resistance biomarkers is made for BRAF/MEK and PDL1 pathways. For example NRas and EGFR bypass can be observed and low IFNg signatures can be observed with resistance. A more complete review of the resistance mechanisms and their biomarkers would better distinguish this review from the many others on targeted therapy in melanoma. Otherwise the content presented i good but somewhat incomplete.

Author Response

Response to Reviewer 1 Comments

The authors review the molecular basis of malignant melanoma, focusing on BRAF/MEK and PDL1 pathways of therapeutic interest. Overlap with metabolic and protein biomarkers is made. The manuscript would be greatly improved if additional information on resistance biomarkers is made for BRAF/MEK and PDL1 pathways. For example NRas and EGFR bypass can be observed and low IFNg signatures can be observed with resistance. A more complete review of the resistance mechanisms and their biomarkers would better distinguish this review from the many others on targeted therapy in melanoma. Otherwise the content presented i good but somewhat incomplete.

Response: Thank you for very critical comments. We agree with the reviewer’s valuable point about resistance biomarkers regarding the BRAF/MEK and PDL1 pathways. We have revised the text accordingly, and examples from NRAS signaling [1] (Page 11, Line 366) and bypass activation of MAPK through receptor tyrosine kinases (RTKs, e.g., EGFR) [2,3] (Page 11, Line 367) have been added, as important mechanisms of resistance in melanoma.

In addition, resistance after treatment with checkpoint inhibitors has been reported including primary resistance to PD-1/PD-L1 or CTLA-4 from the lack of interferon gamma (IFNγ) signaling in the tumors [4-6] (Page 11, Line 373-375). As per the reviewer’s valuable comments, we added recent studies and references and revised accordingly.

[1] Nazarian, R.; Shi, H.; Wang, Q.; Kong, X.; Koya, R.C.; Lee, H.; Chen, Z.; Lee, M.K.; Attar, N.; Sazegar, H.; et al. Melanomas acquire resistance to B-RAF(V600E) inhibition by RTK or N-RAS upregulation. Nature 2010, 468, 973–977.

[2] Corcoran, R.B.; Ebi, H.; Turke, A.B.; Coffee, E.M.; Nishino, M.; Cogdill, A.P.; Brown, R.D.; Pelle, P. Della; Dias-Santagata, D.; Hung, K.E.; et al. EGFR-mediated reactivation of MAPK signaling contributes to insensitivity of BRAF-mutant colorectal cancers to RAF inhibition with vemurafenib. Cancer Discov. 2012, 2, 227–235.

[3] Jenkins, R.W.; Barbie, D.A. Refining Targeted Therapy Opportunities for BRAF-Mutant Melanoma. 2017.

[4] Benci, J.L.; Xu, B.; Qiu, Y.; Wu, T.J.; Dada, H.; Twyman-Saint Victor, C.; Cucolo, L.; Lee, D.S.M.; Pauken, K.E.; Huang, A.C.; et al. Tumor Interferon Signaling Regulates a Multigenic Resistance Program to Immune Checkpoint Blockade. Cell 2016, 167, 1540-1554.e12.

[5] Nowicki, T.S.; Hu-Lieskovan, S.; Ribas, A. Mechanisms of Resistance to PD-1 and PD-L1 Blockade. Cancer J. (United States) 2018, 24, 47–53.

[6] Gao, J.; Shi, L.Z.; Zhao, H.; Chen, J.; Xiong, L.; He, Q.; Chen, T.; Roszik, J.; Bernatchez, C.; Woodman, S.E.; et al. Loss of IFN-γ Pathway Genes in Tumor Cells as a Mechanism of Resistance to Anti-CTLA-4 Therapy. Cell 2016, 167, 397-404.e9.

Reviewer 2 Report

  1. Concerning figure 1, I would recommend to remove it, since it is not really necessary, is too general.
  2. Introduction is also too general.
  3. Figure 2 and 3 have different bar size.
  4. I would suggest to include a paragraph why chemotherapy is seldom used any longer v/s immunotherapy in MM. 

Author Response

Response to Reviewer 2 Comments

Reviewer #2:

#1. Concerning figure 1, I would recommend to remove it, since it is not really necessary, is too general.

#2. Introduction is also too general.

Response: Thank you for your valuable recommendation. We agree with the reviewer’s valuable point about Figure 1 and Introduction.

Regarding Figure 1, we have revised, and added detail parts into the illustration, helping the reader to get a clear view upfront what the manuscript is capturing and reporting on., by expalining the holistic view of melanoma signatures for disease presentation. From melanoma tissues to molecular level of human body, specific molecular features, which are including genome, transcriptome, proteome, and etc., are directly connected with melanoma disease signatures for disease presentation. Based on these features, upregulated metastatic mechanisms are driving to uncontrolled proliferation and advanced metastasis showing high heterogeneous cells in MM (Page 2, Line 68-73). Additionally, in order to overcome the generality of the context within the Introduction, we added our mission statement from the European CancerMoonshot Center ‘we conduct cancer research to end cancer as we know it, and to help benefit society as a whole’ (Page 2, Line 67-68). 

#3. Figure 2 and 3 have different bar size.

Response: Thank you for pointing out these errors in Figure 2 and 3. The errors are corrected in the revised manuscript accordingly.

#4. I would suggest to include a paragraph why chemotherapy is seldom used any longer v/s immunotherapy in MM.

Response: Thank you for your valuable suggestion. In contrast to many other cancer types chemotherapy is now rarely used as first line treatment for malignant melanoma due to poor efficacy, side effects and to the fact that more effective treatment options now are available [1] (Page 10, Line 316-318).

Further, we also added recent and promising immunotherapies with its references. Lately, engineered oncolytic viruses have been shown to be effective for local and systemic immunotherapies in cancer. A few years ago, the first oncolytic virus was approved by the FDA for melanoma immunotherapy [2-5]. This therapy is used not only for non-resectable skin and lymph node lesions but also for melanoma tissue elsewhere [6]. Recurrent, unrespectable stage IIIB-IVM1a melanoma patients have been treated with talimogene laherparepvec (T-VEC), the oncolytic virus, which is the first approved therapy [2]. Accordingly, for explanation of clear meaning on this, we revised the previous description of this part and added references (Page 12, Line 415-420).

[1] Mattia, G.; Puglisi, R.; Ascione, B.; Malorni, W.; Carè, A.; Matarrese, P. Cell death-based treatments of melanoma:conventional treatments and new therapeutic strategies review-Article. Cell Death Dis. 2018, 9.

[2] Pol, J.; Kroemer, G.; Galluzzi, L. First oncolytic virus approved for melanoma immunotherapy. Oncoimmunology 2016, 5, e1115641.

[3] Kaufman, H.L.; Kohlhapp, F.J.; Zloza, A. Oncolytic viruses: A new class of immunotherapy drugs. Nat. Rev. Drug Discov. 2015, 14, 642–662.

[4] Reale, A.; Vitiello, A.; Conciatori, V.; Parolin, C.; Calistri, A.; Palù, G. Perspectives on immunotherapy via oncolytic viruses. Infect. Agent. Cancer 2019, 14, 5.

[5] Lugowska, I.; Teterycz, P.; Rutkowski, P. Immunotherapy of melanoma. Wspolczesna Onkol. 2017, 2, 61–67.

[6] Rothermel, L.D.; Zager, J.S. Engineered oncolytic viruses to treat melanoma: where are we now and what comes next? Expert Opin. Biol. Ther. 2018, 18, 1199–1207.

Round 2

Reviewer 1 Report

Revisions are fine. Acceptable in present form.